# Improved Graph Laplacian via Geometric Consistency

**Dominique C. Perrault-Joncas**
Google, Inc.
dominiquep@google.com

**Marina Meilă**
Department of Statistics
University of Washington
mmp2@uw.edu

**James McQueen**
Amazon
jmcq@amazon.com

## Abstract

In all manifold learning algorithms and tasks setting the kernel bandwidth $\epsilon$ used construct the graph Laplacian is critical. We address this problem by choosing a quality criterion for the Laplacian, that measures its ability to preserve the geometry of the data. For this, we exploit the connection between manifold geometry, represented by the Riemannian metric, and the Laplace-Beltrami operator. Experiments show that this principled approach is effective and robust.

## 1 Introduction

Manifold learning and manifold regularization are popular tools for dimensionality reduction and clustering [1, 2], as well as for semi-supervised learning [3, 4, 5, 6] and modeling with Gaussian Processes [7]. Whatever the task, a manifold learning method requires the user to provide an external parameter, called "bandwidth" or "scale" $\epsilon$, that defines the size of the local neighborhood.

More formally put, a common challenge in semi-supervised and unsupervised manifold learning lies in obtaining a "good" graph Laplacian estimator $L$. We focus on the practical problem of optimizing the parameters used to construct $L$ and, in particular, $\epsilon$. As we see empirically, since the Laplace-Beltrami operator on a manifold is intimately related to the geometry of the manifold, our estimator for $\epsilon$ has advantages even in methods that do not explicitly depend on $L$.

In manifold learning, there has been sustained interest for determining the asymptotic properties of $L$ [8, 9, 10, 11]. The most relevant is [12], which derives the optimal rate for $\epsilon$ w.r.t. the sample size $N$

$$\epsilon^2 = C(\mathcal{M})N^{-\frac{1}{3+d/2}}, \tag{1}$$

with $d$ denoting the intrinsic dimension of the data manifold $\mathcal{M}$. The problem is that $C(\mathcal{M})$ is a constant that depends on the yet unknown data manifold, so it is rarely known in practice.

Considerably fewer studies have focused on the parameters used to construct $L$ in a finite sample problem. A common approach is to "tune" parameters by cross-validation in the semi-supervised context. However, in an unsupervised problem like non-linear dimensionality reduction, there is no context in which to apply cross-validation. While several approaches [13, 14, 15, 16] may yield a usable parameter, they generally do not aim to improve $L$ *per se* and offer no geometry-based justification for its selection.

In this paper, we present a new, geometrically inspired approach to selecting the bandwidth parameter $\epsilon$ of $L$ for a given data set. Under the data manifold hypothesis, the *Laplace-Beltrami* operator $\Delta_{\mathcal{M}}$ of the data manifold $\mathcal{M}$ contains all the intrinsic geometry of $\mathcal{M}$. We set out to exploit this fact by comparing the geometry induced by the graph Laplacian $L$ with the local data geometry and choose the value of $\epsilon$ for which these two are closest.

## 2  Background: Heat Kernel, Laplacian and Geometry

Our paper builds on two previous sets of results: 1) the construction of $L$ that is consistent for $\Delta_{\mathcal{M}}$ when the sample size $N \to \infty$ under the data manifold hypothesis (see [17]); and 2) the relationship between $\Delta_{\mathcal{M}}$ and the Riemannian metric $g$ on a manifold, as well as the estimation of $g$ (see [18]).

**Construction of the graph Laplacian.** Several methods methods to construct $L$ have been suggested (see [10, 11]). The one we present, due to [17], guarantees that, if the data are sampled from a manifold $\mathcal{M}$, $L$ converges to $\Delta_{\mathcal{M}}$:

Given a set of points $\mathcal{D} = \{x_1, \ldots, x_N\}$ in high-dimensional Euclidean space $\mathbb{R}^r$, construct a weighted graph $\mathcal{G} = (\mathcal{D}, W)$ over them, with $W = [w_{ij}]_{ij=1:N}$. The weight $w_{ij}$ between $x_i$ and $x_j$ is the *heat kernel* [1]

$$W_{ij} \equiv w_\epsilon(x_i, x_j) = \exp\left(||x_i - x_j||_2^2 / \epsilon^2\right),\tag{2}$$

with $\epsilon$ a *bandwidth* parameter fixed by the user. Next, construct $L = [L_{ij}]_{ij}$ of $\mathcal{G}$ by

$$t_i = \sum_j W_{ij}, \quad W'_{ij} = \frac{W_{ij}}{t_i t_j}, \quad t'_i = \sum_j W'_{ij}, \text{ and } L_{ij} = \sum_j \frac{W'_{ij}}{t'_j}.\tag{3}$$

Equation (3) represents the discrete versions of the renormalized Laplacian construction from [17]. Note that $t_i, t'_i, W', L$ all depend on the bandwidth $\epsilon$ via the heat kernel.

**Estimation of the Riemannian metric.** We follow [18] in this step. A *Riemannian manifold* $(\mathcal{M}, g)$ is a *smooth manifold* $\mathcal{M}$ endowed with a *Riemannian metric* $g$; the metric $g$ at point $p \in \mathcal{M}$ is a scalar product over the vectors in $\mathcal{T}_p\mathcal{M}$, the *tangent subspace* of $\mathcal{M}$ at $p$. In any coordinate representation of $\mathcal{M}$, $g_p \equiv G(p)$ - the Riemannian metric at $p$ - represents a positive definite matrix[1] of dimension $d$ equal to the *intrinsic dimension* of $\mathcal{M}$. We say that the metric $g$ encodes the geometry of $\mathcal{M}$ because $g$ determines the *volume element* for any integration over $\mathcal{M}$ by $\sqrt{\det G(x)}dx$, and the *line element* for computing distances along a curve $x(t) \subset \mathcal{M}$, by $\sqrt{\left(\frac{dx}{dt}\right)^T G(x) \frac{dx}{dt}}$.

If we assume that the data we observe (in $\mathbb{R}^r$) lies on a manifold, then under rotation of the original coordinates, the metric $G(p)$ is the unit matrix of dimension $d$ padded with zeros up to dimension $r$. When the data is mapped to another coordinate system - for instance by a manifold learning algorithm that performs non-linear dimension reduction - the matrix $G(p)$ changes with the coordinates to reflect the distortion induced by the mapping (see [18] for more details).

**Proposition 2.1** *Let $x$ denote local coordinate functions of a smooth Riemannian manifold $(\mathcal{M}, g)$ of dimension $d$ and $\Delta_{\mathcal{M}}$ the Laplace-Beltrami operator defined on $\mathcal{M}$. Then, $H(p) = (G(p))^{-1}$ the (matrix) inverse of the Riemannian metric at point $p$, is given by*

$$(H(p))_{kj} = \tfrac{1}{2}\Delta_{\mathcal{M}}\left(x^k - x^k(p)\right)\left(x^j - x^j(p)\right)|_{x=x(p)} \quad \text{with } i, j = 1, \ldots, d.\tag{4}$$

Note that the inverse matrices $H(p), p \in \mathcal{M}$, being symmetric and positive definite, also defines a metric $h$ called the *cometric* on $\mathcal{M}$. Proposition 2.1 says that the cometric is given by applying the $\Delta_{\mathcal{M}}$ operator to the function $\phi_{kj} = \left(x^k - x^k(p)\right)\left(x^j - x^j(p)\right)$, where $x^k, x^j$ denote coordinates $k, j$ seen as functions on $\mathcal{M}$. A converse theorem [19] states that $g$ (or $h$) uniquely determines $\Delta_{\mathcal{M}}$. Proposition 2.1 provides a way to estimate $h$ and $g$ from data. Algorithm 1, adapted from [18], implements (4).

## 3  A Quality Measure for $L$

Our approach can be simply stated: the "best" value for $\epsilon$ is the value for which the corresponding $L$ of (3) best captures the original data geometry. For this we must: (1) estimate the geometry $g$ or $h$

**Algorithm 1** Riemannian Metric($X, i, L, pow \in \{-1, 1\}$)

---

**Input:** $N \times d$ design matrix $X$, $i$ index in data set, Laplacian $L$, binary variable $pow$
**for** $k = 1 \to d, l = 1 \to d$ **do**
$\quad H_{k,l} \leftarrow \sum_{j=1}^{N} L_{ij} (X_{jk} - X_{ik})(X_{jl} - X_{il})$
**end for**
**return** $H^{pow}$ (i.e. $H$ if $pow = 1$ and $H^{-1}$ if $pow = -1$)

---

from $L$ (this is achieved by RiemannianMetric()); (2) find an independent way to estimate the data geometry, locally (this is done in Sections 3.2 and 3.1); (3) define a measure of agreement between the two (Section 3.3).

## 3.1  The Geometric Consistency Idea and $g^{target}$

There is a natural way to estimate the geometry of the data without the use of $L$. We consider the canonical embedding of the data in the ambient space $\mathbb{R}^r$ for which the geometry is trivially known. This provides a target $g^{target}$; we tune the scale of the Laplacian so that the $g$ calculated from Proposition 2.1 matches this target. Hence, we choose $\epsilon$ to maximize *consistency* with the geometry of the data. We denote the inherited metric by $g_{\mathbb{R}^r}|_{T\mathcal{M}}$, which stands for the restriction of the natural metric of the ambient space $\mathbb{R}^r$ to the *tangent bundle* $T\mathcal{M}$ of the manifold $\mathcal{M}$. We tune the parameters of the graph Laplacian $L$ so as to enforce (a coordinate expression of) the identity

$$g_p(\epsilon) = g^{target}, \quad \text{with } g^{target} = g_{\mathbb{R}^r}|_{T_p\mathcal{M}} \ \forall p \in \mathcal{M} . \tag{5}$$

In the above, the l.h.s. will be the metric implied from the Laplacian via Proposition 2.1, and the r.h.s is the metric induced by $\mathbb{R}^r$. Mathematically speaking, (5) is necessary and sufficient for finding the "correct" Laplacian. The next section describes how to obtain the r.h.s. from a finite sample $\mathcal{D}$. Then, to optimize the graph Laplacian we estimate $g$ from $L$ as prescribed by Proposition 2.1 and compare with $g_{\mathbb{R}^r}|_{T_p\mathcal{M}}$ numerically. We call this approach *geometric consistency (GC)*. The GC method is not limited to the choice of $\epsilon$, but can be applied to any other parameter required for the Laplacian.

## 3.2  Robust Estimation of $g^{target}$ for a finite sample

**First idea: estimate tangent subspace**  We use the simple fact, implied by Section 3.1, that projecting the data onto $T_p\mathcal{M}$ preserves the metric locally around $p$. Hence, $G^{target} = I_d$ in the projected data. Moreover, projecting on any direction in $T_p\mathcal{M}$ does not change the metric in that direction. This remark allows us to work with small matrices (of at most $d \times d$ instead of $r \times r$) and to avoid the problem of estimating $d$, the intrinsic dimension of the data manifold.

Specifically, we evaluate the tangent subspace around each sampled point $x_i$ using *weighted (local) Principal Component Analysis (wPCA)* and then express $g_{\mathbb{R}^r}|_{T_p\mathcal{M}}$ directly in the resulting low-dimensional subspace as the unit matrix $I_d$. The tangent subspace also serves to define a local coordinate chart, which is passed as input to Algorithm 1 which computes $H(x_i), G(x_i)$ in these coordinates. For computing $T_{x_i}\mathcal{M}$, by wPCA, we choose weights defined by the heat kernel (2), centered around $x_i$, with same bandwidth $\epsilon$ as for computing $L$. This approach is similar to sample-wise weighted PCA of [20], with one important requirements: the weights must decay rapidly away from $x_i$ so that only points close $x_i$ are used to estimate $T_{x_i}\mathcal{M}$. This is satisfied by the weighted recentered design matrix $Z$, where $Z_{i:}$, row $i$ of $Z$, is given by:

$$Z_{i:} = W_{ij}(x_i - \bar{x}) / \left( \sum_{j'=1}^{N} W_{ij'} \right), \quad \text{with } \bar{x} = \left( \sum_{j=1}^{N} W_{ij}x_j \right) / \left( \sum_{j'=1}^{N} W_{ij'} \right) . \tag{6}$$

[21] proves that the wPCA using the heat kernel, and equating the PCA and heat kernel bandwidths as we do, yields a consistent estimator of $T_{x_i}\mathcal{M}$. This is implemented in Algorithm 2.

In summary, to instantiate equation (5) at point $x_i \in \mathcal{D}$, one must (i) construct row $i$ of the graph Laplacian by (3); (ii) perform Algorithm 2 to obtain $Y$; (iii) apply Algorithm 1 to $Y$ to obtain $G(x_i) \in \mathbb{R}^{d \times d}$; (iv) this matrix is then compared with $I_d$, which represents the r.h.s. of (5).

---

**Algorithm 2** Tangent Subspace Projection$(X, w, d')$

---

**Input:** $N \times r$ design matrix $X$, weight vector $w$, working dimension $d'$
Compute $Z$ using (6)
$[V, \Lambda] \leftarrow \mathrm{eig}(Z^T Z, d')$    (i.e.$d'$-SVD of $Z$)
Center $X$ around $\bar{x}$ from (6)
$Y \leftarrow X V_{:,1:d'}$ (Project $X$ on $d'$ principal subspace)
**return** Y

---

**Second idea: project onto tangent directions**    We now take this approach a few steps further in terms of improving its robustness with minimal sacrifice to its theoretical grounding. In particular, we perform both Algorithm 2 and Algorithm 1 in $d'$ dimensions, with $d' < d$ (and typically $d' = 1$). This makes the algorithm faster, and make the computed metrics $G(x_i), H(x_i)$ both more stable numerically and more robust to possible noise in the data[2]. Proposition 3.1 shows that the resulting method remains theoretically sound.

**Proposition 3.1** *Let $X$, $Y$, $Z$, $V$, $W_{:i}$, $H$, and $d \geq 1$ represent the quantities in Algorithms 1 and 2; assume that the columns of $V$ are sorted in decreasing order of the singular values, and that the rows and columns of $H$ are sorted according to the same order. Now denote by $Y'$, $V'$, $H'$ the quantitities computed by Algorithms 1 and 2 for the same $X$, $W_{:i}$ but with $d \leftarrow d' = 1$. Then,*

$$V' = V_{:1} \in \mathbb{R}^{r \times 1} \ \ Y' = Y_{:1} \in \mathbb{R}^{N \times 1} \ \ H' = H_{11} \in \mathbb{R}. \tag{7}$$

The proof of this result is straightforward and omitted for brevity. It is easy to see that Proposition 3.1 generalizes immediately to any $1 \leq d' < d$. In other words, by using $d' < d$, we will be projecting the data on a proper subspace of $T_{x_i}\mathcal{M}$ - namely, the subspace of least curvature [22]. The cometric $H'$ of this projection is the principal submatrix of order $d'$ of $H$, i.e. $H_{11}$ if $d' = 1$.

**Third idea: use $h$ instead of $g$**    Relation (5) is trivially satisfied by the cometrics of $g$ and $g^{target}$ (the latter being $H^{target} = I_d$). Hence, inverting $H$ in Algorithm 1 is not necessary, and we will use the cometric $h$ in place of $g$ by default. This saves time and increases numerical stability.

### 3.3    Measuring the Distortion

For a finite sample, we cannot expect (5) to hold exactly, and so we need to define a distortion between the two metrics to evaluate how well they agree. We propose the *distortion*

$$D = \frac{1}{N} \sum_{i=1}^{N} ||H(x_i) - I_d|| \tag{8}$$

where $||A|| = \lambda_{max}(A)$ is the matrix spectral norm. Thus $D$ measures the average distance of $H$ from the unit matrix over the data set. For a "good" Laplacian, the distortion $D$ should be minimal:

$$\hat{\epsilon} = \mathrm{argmin}_\epsilon D. \tag{9}$$

The choice of norm in (8) is not arbitrary. Riemannian metrics are order 2 tensors or $T\mathcal{M}$ hence the expression of $D$ is the discrete version of $D_{g_0}(g_1, g_2) = \int_{\mathcal{M}} ||g_1 - g_2||_{g_0} dV_{g_0}$, with $||g||_{g_0}\big|_p = \sup_{u,v \in \mathcal{T}_p\mathcal{M}\setminus\{0\}} \frac{<u,v>_{g_p}}{<u,v>_{g_{0p}}}$, representing the *tensor norm* of $g_p$ on $\mathcal{T}_p\mathcal{M}$ with respect to the Riemannian metric $g_{0p}$. Now, (8) follows when $g_0, g_1, g_2$ are replaced by $I$, $I$ and $H$, respectively.

With (9), we have established a principled criterion for selecting the parameter(s) of the graph Laplacian, by minimizing the distortion between the true geometry and the geometry derived from Proposition 2.1. Practically, we have in (9) a 1D optimization problem with no derivatives, and we can use standard algorithms to find its minimum. $\hat{\epsilon}$.

## 4    Related Work

We have already mentioned the asymptotic result (1) of [12]. Other work in this area [8, 10, 11, 23] provides the rates of change for $\epsilon$ with respect to $N$ to guarantee convergence. These studies are

**Algorithm 3** Compute Distortion$(X, \epsilon, d')$

---

**Input:** $N \times r$ design matrix $X$, $\epsilon$, working dimension $d'$, index set $\mathcal{I} \subseteq \{1, \ldots, N\}$
Compute the heat kernel $W$ by (2) for each pair of points in $X$
Compute the graph Laplacian $L$ from $W$ by (3)
$D \leftarrow 0$
**for** $i \in \mathcal{I}$ **do**
    $Y \leftarrow \text{TangentSubspaceProjection}(X, W_{i,:}, d')$
    $H \leftarrow \text{RiemannianMetric}(Y, L, pow = 1)$
    $D \leftarrow D + ||H - I_{d'}||^2/|\mathcal{I}|$
**end for**
**return** $D$

---

relevant; but they depend on manifold parameters that are usually not known. Recently, an extremely interesting Laplacian "continuous nearest neighbor" consistent construction method was proposed by [24], from a topological perspective. However, this method depends on a smoothness parameter too, and this is estimated by constructing the *persistence diagram* of the data. [25] propose a new, statistical approach for estimating $\epsilon$, which is very promising, but currently can be applied only to un-normalized Laplacian operators. This approach also depends on unknown pparameters $a, b$, which are set heuristically. (By contrast, our method depends only weakly on $d'$, which can be set to 1.)

Among practical methods, the most interesting is that of [14], which estimates $k$, the number of nearest neighbors to use in the construction of the graph Laplacian. This method optimizes $k$ depending on the embedding algorithm used. By contrast, the selection algorithm we propose estimates an *intrinsic* quantity, a scale $\epsilon$ that depends exclusively on the data. Moreover, it is not known when minimizing reconstruction error for a particular method can be optimal, since [26] even in the limit of infinite data, the most embeddings will distort the original geometry. In semi-supervised learning (SSL), one uses Cross-Validation (CV) [5].

Finally, we mention the algorithm proposed in [27] (CLMR). Its goal is to obtain an estimate of the intrinsic dimension of the data; however, a by-product of the algorithm is a range of scales where the tangent space at a data point is well aligned with the principal subspace obtained by a local singular value decomposition. As these are scales at which the manifold looks locally linear, one can reasonably expect that they are also the correct scales at which to approximate differential operators, such as $\Delta_{\mathcal{M}}$. Given this, we implement the method and compare it to our own results.

From the computational point of view, all methods described above explore exhaustively a range of $\epsilon$ values. GC and CLMR only require local PCA at a subset of the data points (with $d' < d$ components for GC, $d' >> d$ for CLMR); whereas CV, and [14] require respectively running a SSL algorithm, or an embedding algorithm, for each $\epsilon$. In relation to these, GC is by far the most efficient computationally. [3]

## 5 Experimental Results

**Synthethic Data.** We experimented with estimating the bandwidth $\hat{\epsilon}$ on data sampled from two known manifolds, the two-dimensional `hourglass` and `dome` manifolds of Figure 1. We sampled points uniformly from these, adding 10 "noise" dimensions and Gaussian noise $\mathcal{N}(0, \sigma^2)$ resulting in $r = 13$ dimensions.

The range of $\epsilon$ values was delimited by $\epsilon_{min}$ and $\epsilon_{max}$. We set $\epsilon_{max}$ to the average of $||x_i - x_j||^2$ over all point pairs and $\epsilon_{min}$ to the limit in which the heat kernel $W$ becomes approximately equal to the unit matrix; this is tested by $\max_j(\sum_i W_{ij}) - 1 < \gamma^4$ for $\gamma \approx 10^{-4}$. This range spans about two orders of magnitude in the data we considered, and was searched by a logarithmic grid with approximately 20 points. We saved computatation time by evaluating all pointwise quantities ($\hat{D}$, local SVD) on a random sample of size $N' = 200$ of each data set. We replicated each experiment on 10 independent samples.

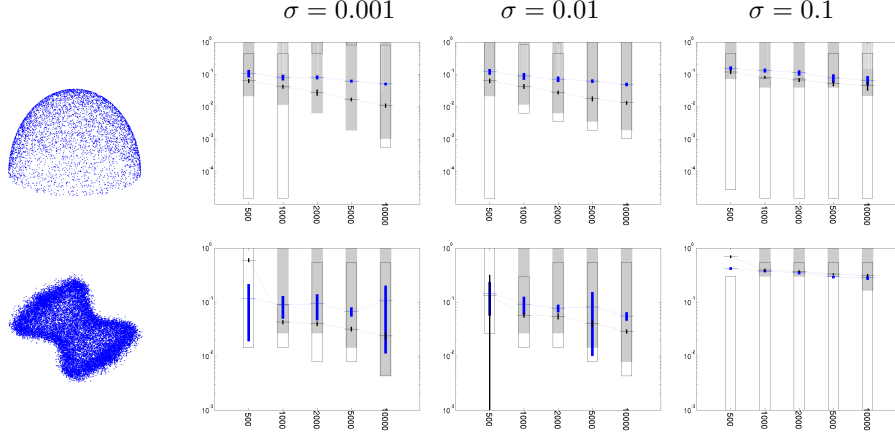

Figure 1: Estimates $\hat{\epsilon}$ (mean and standard deviation over 10 runs) on the `dome` and `hourglass` data, vs sample sizes $N$ for various noise levels $\sigma$; $d' = 2$ is in black and $d' = 1$ in blue. In the background, we also show as gray rectangles, for each $N, \sigma$ the intervals in the $\epsilon$ range where the eigengaps of local SVD indicate the true dimension, and, as unfilled rectangles, the estimates proposed by CLMR [27] for these intervals. The variance of $\hat{\epsilon}$ observed is due to randomness in the subsample $N'$ used to evaluate the distortion. Our $\hat{\epsilon}$ always falls in the true interval (when this exists), and have are less variable and more accurate than the CLMR intervals.

**Reconstruction of manifold w.r.t. gold standard** These results (relegated to the Supplement) are uniformly very positive, and show that `GC` achieves its most explicit goal, even in the presence of noise. In the remainder, we illustrate the versatility of our method on on other tasks. **Effects of $d'$, noise and $N$.** The estimated $\epsilon$ are presented in Figure 1. Let $\hat{\epsilon}_{d'}$ denote the estimate obtained for a given $d' \leq d$. We note that when $d_1 < d_2$, typically $\hat{\epsilon}_{d_1} > \hat{\epsilon}_{d_2}$, but the values are of the same order (a ratio of about 2 in the synthetic experiments). The explanation is that, chosing $d' < d$ directions in the tangent subspace will select a subspace aligned with the "least curvature" directions of the manifold, if any exist, or with the "least noise" in the random sample. In these directions, the data will tolerate more smoothing, which results in larger $\hat{\epsilon}$. The optimal $\epsilon$ decreases with $N$ and grows with the noise levels, reflecting the balance it must find between variance and bias. Note that for the `hourglass` data, the highest noise level of $\sigma = 0.1$ is an extreme case, where the original manifold is almost drowned in the 13-dimensional noise. Hence, $\epsilon$ is not only commensurately larger, but also stable between the two dimensions and runs. This reflects the fact that $\epsilon$ captures the noise dimension, and its values are indeed just below the noise amplitude of $0.1\sqrt{13}$. The `dome` data set exhibits the same properties discussed previously, showing that our method is effective even for manifolds with border.

**Semi-supervised Learning (SSL) with Real Data.** In this set of experiments, the task is classification on the benchmark SSL data sets proposed by [28]. This was done by least-square classification, similarly to [5], after choosing the optimal bandwidth by one of the methods below.

TE *Minimize Test Error*, i.e. "cheat" in an attempt to get an estimate of the "ground truth".

CV *Cross-validation* We split the training set (consisting of 100 points in all data sets) into two equal groups;[5] we minimize the highly non-smooth CV classification error by simulated annealing.

Rec *Minimize the reconstruction error* We cannot use the method of [14] directly, as it requires an embedding, so we minimize reconstruction error based on the heat kernel weights w.r.t. $\epsilon$ (this is reminiscent of LLE [29]): $\mathcal{R}(\epsilon) = \sum_{i=1}^{n} \left|\left| x_i - \sum_{j \neq i} \frac{W_{ij}}{\sum_{l \neq i} W_{ij}} x_j \right|\right|^2$

Our method is denoted `GC` for *Geometric Consistency*; we evaluate straighforward `GC`, that uses the cometric $H$ and a variant that includes the matrix inversion in Algorithm 1 denoted $\texttt{GC}^{-1}$.

| | TE | CV | Rec | GC$^{-1}$ | GC |
|---|---|---|---|---|---|
| Digit1 | 0.67±0.08 [0.57, 0.78] | 0.80±0.45 [0.47, 1.99] | 0.64 | 0.74 | 0.74 |
| USPS | 1.24±0.15 [1.04, 1.59] | 1.25±0.86 [0.50, 3.20] | 1.68 | 2.42 | 1.10 |
| COIL | 49.79±6.61 [42.82, 60.36] | 69.65±31.16 [50.55, 148.96] | 78.37 | 216.95 | 116.38 |
| BCI | 3.4±3.1 [1.2, 8.9] | 3.2±2.5 [1.2, 8.2] | 3.31 | 3.19 | 5.61 |
| g241c | 8.3± 2.5 [6.3, 14.6] | 8.8±3.3 [4.4, 14.9] | 3.79 | 7.37 | 7.38 |
| g241d | 5.7± 0.24 [5.6, 6.3] | 6.4±1.15 [4.3, 8.2] | 3.77 | 7.35 | 7.36 |

Table 1: Estimates of $\epsilon$ by methods presented for the six SSL data sets used, as well as TE. For TE and CV, which depend on the training/test splits, we report the average, its standard error, and range (in brackets below) over the 12 splits.

| | CV | Rec | GC$^{-1}$ | GC |
|---|---|---|---|---|
| Digit1 | 3.32 | 2.16 | 2.11 | 2.11 |
| USPS | 5.18 | 4.83 | 12.00 | 3.89 |
| COIL | 7.02 | 8.03 | 16.31 | 8.81 |
| BCI | 49.22 | 49.17 | 50.25 | 48.67 |
| g241c | 13.31 | 23.93 | 12.77 | 12.77 |
| g241d | 8.67 | 18.39 | 8.76 | 8.76 |

| | | $d'$=1 | $d'$=2 | $d'$=3 |
|---|---|---|---|---|
| Digit1 | GC$^{-1}$ | 0.743 | 0.293 | 0.305 |
| | GC | 0.744 | 0.767 | 0.781 |
| USPS | GC$^{-1}$ | 2.42 | 2.31 | 3.88 |
| | GC | 1.10 | 1.16 | 1.18 |
| COIL | GC$^{-1}$ | 116 | 87.4 | 128 |
| | GC | 187 | 179 | 187 |
| BCI | GC$^{-1}$ | 3.32 | 3.48 | 3.65 |
| | GC | 5.34 | 5.34 | 5.34 |
| g241c | GC$^{-1}$ | 7.38 | 7.38 | 7.38 |
| | GC | 7.38 | 9.83 | 9.37 |
| g241d | GC$^{-1}$ | 7.35 | 7.35 | 7.35 |
| | GC | 7.35 | 9.33 | 9.78 |

Table 2: *Left panel*: Percent classification error for the six SSL data sets using the four $\epsilon$ estimation methods described. *Right panel*: $\epsilon$ obtained for the six datasets using various $d'$ values with GC and GC$^{-1}$ . $\hat{\epsilon}$ was computed for $d$=5 for Digit1, as it is known to have an intrinsic dimension of 5, and found to be 1.162 with GC and 0.797 with GC$^{-1}$ .

Across all methods and data sets, the estimate of $\epsilon$ closer to the values determined by TE lead to better classification error, see Table 2. For five of the six data sets[6], GC-based methods outperformed CV, and were 2 to 6 times faster to compute. This is in spite of the fact that GC does not use label information, and is not aimed at reducing the classification error, while CV does. Further, the CV estimates of $\epsilon$ are highly variable, suggesting that CV tends to overfit to the training data.

**Effect of Dimension** $d'$**.** Table 2 shows how changing the dimension $d'$ alters our estimate of $\epsilon$. We see that the $\hat{\epsilon}$ for different $d'$ values are close, even though we search over a range of two orders of magnitude. Even for g241c and g241d, which were constructed so as to not satisfy the manifold hypothesis, our method does reasonably well at estimating $\epsilon$. That is, our method finds the $\hat{\epsilon}$ for which the Laplacian encodes the geometry of the data set irrespective of whether or not that geometry is lower-dimensional. Overall, we have found that using $d' = 1$ is most stable, and that adding more dimensions introduces more numerical problems: it becomes more difficult to optimize the distortion as in (9), as the minimum becomes shallower. In our experience, this is due to the increase in variance associated with adding more dimensions.

Using one dimension probably works well because the wPCA selects the dimension that explains the most variance and hence is the closest to linear over the scale considered. Subsequently, the wPCA moves to incrementally "shorter" or less linear dimensions, leading to more variance in the estimate of the tangent subspace (more evidence for this in the Supplement).

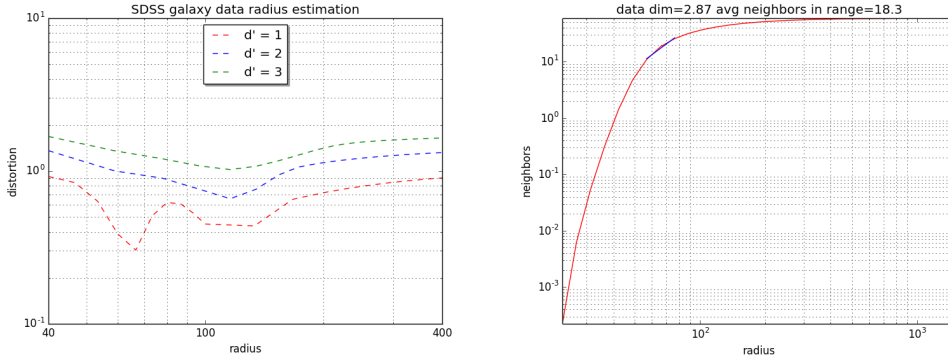

Figure 2: Bandwidth Estimation For Galaxy Spectra Data. Left: GC results for $d' = 1$ ($d' = 2, 3$ are also shown); we chose radius = 66 the minimum of $D$ for $d = 1'$. Right: A log-log plot of radius versus average number of neighbors within this radius. The region in blue includes radius = 66 and indicates dimension $d = 3$. In the code $\epsilon = \text{radius}/3$, hence we use $\epsilon = 22$.

**Embedding spectra of galaxies** (Details of this experiment are in the Supplement.) For these data in $r = 3750$ dimensions, with $N = 650,000$, the goal was to obtain a smooth, low dimensional embedding. The intrinsic dimension $d$ is unknown, CV cannot be applied, and it is impractical to construct multiple embeddings for large $N$. Hence, we used the GC method with $d' = 1, 2, 3$ and $N' = |\mathcal{I}| = 200$. We compare the $\hat{\epsilon}$'s obtained with a heuristic based on the scaling of the neighborhood sizes [30] with the radius, which relates $\epsilon, d$ and $N$ (Figure 2). Remarkably, both methods yield the same $\epsilon$, see the Supplement for evidence that the resulting embedding is smooth.

## 6    Discussion

In manifold learning, supervised and unsupervised, estimating the graph versions of Laplacian-type operators is a fundamental task. We have provided a principled method for selecting the parameters of such operators, and have applied it to the selection of the bandwidth/scale parameter $\epsilon$. Moreover, our method can be used to optimize any other parameters used in the graph Laplacian; for example, $k$ in the $k$-nearest neighbors graph, or - more interestingly - the renormalization parameter $\lambda$ [17] of the kernel. The latter is theoretically equal to 1, but it is possible that it may differ from 1 in the finite $N$ regime. In general, for finite $N$, a small departure from the asymptotic prescriptions may be beneficial - and a data-driven method such as ours can deliver this benefit.

By imposing geometric self-consistency, our method estimates an *intrinsic quantity* of the data. GC is also fully unsupervised, aiming to optimize a (lossy) representation of the data, rather than a particular task. This is an efficiency if the data is used in an unsupervised mode, or if it is used in many different subsequent tasks. Of course, one cannot expect an unsupervised method to always be superior to a task-dependent one. Yet, GC has shown to be competitive and sometimes superior in experiments with the widely accepted CV. Besides the experimental validation, there are other reasons to consider an unsupervised method like GC in a supervised task: (1) the labeled data is scarce, so $\hat{\epsilon}$ will have high variance, (2) the CV cost function is highly non-smooth while $D$ is much smoother, and (3) when there is more than one parameter to optimize, difficulties (1) and (2) become much more severe.

Our algorithm requires minimal prior knowledge. In particular, it *does not* require exact knowledge of the intrinsic dimension $d$, since it can work satisfactorily with $d' = 1$ in many cases.

An interesting problem that is outside the scope of our paper is the question of whether $\epsilon$ needs to vary over $\mathcal{M}$. This is a question/challenge facing not just GC, but any method for setting the scale, unsupervised or supervised. Asymptotically, a uniform $\epsilon$ is sufficient. Practically, however, we believe that allowing $\epsilon$ to vary may be beneficial. In this respect, the GC method, which simply evaluates the overall result, can be seamlessly adapted to work with any user-selected spatially-variable $\epsilon$, by appropriately changing (2) or sub-sampling $\mathcal{D}$ when calculating $D$.

## Footnotes

[1]This paper contains mathematical objects like $\mathcal{M}$, $g$ and $\Delta$, and computable objects like a data point $x$, and the graph Laplacian $L$. The Riemannian metric at a point belongs to both categories, so it will sometimes be denoted $g_p, g_{x_i}$ and sometimes $G(p), G(x_i)$, depending on whether we refer to its mathematical or algorithmic aspects (or, more formally, whether the expression is *coordinate free* or in a given set of coordinates). This also holds for the cometric $h$, defined in Proposition 2.1.

[2]We know from matrix perturbation theory that noise affects the $d$-th principal vector increasingly with $d$.

[3]In addition, these operations being local, they can be further parallelized or accelerated in the usual ways.

[4]Guaranteeing that all eigenvalues of $W$ are less than $\gamma$ away from 1.

[5]In other words, we do 2-fold CV. We also tried 20-fold and 5-fold CV, with no significant difference.

[6]In the COIL data set, despite their variability, CV estimates still outperformed the GC-based methods. This is the only data set constructed from a collection of manifolds - in this case, 24 one-dimensional image rotations. As such, one would expect that there would be more than one natural length scale.

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
