[Supplementary Material]

# SUPPLEMENT FOR: Improved Graph Laplacian via Geometric Self-Consistency

**Dominique C. Perrault-Joncas**
Google, Inc.
dominiquep@google.com

**Marina Meilă**
Department of Statistics
University of Washington
mmp2@uw.edu

**James McQueen**
Amazon
jmcq@amazon.com

## 1  Additional experimental results

### 1.1  Example displaying the cost function for chosing $\epsilon$

This example uses semi-supervised learning (SSL) dataset g241d.

Figure 1 (a) shows the distortion $D$ that our algorithm minimizes to find the optimal $\epsilon$ for the given data set. Figure 1 (b) illustrates the range of $\epsilon$ chosen by the CLMR method. The CLMR range is $[\epsilon_1, \epsilon_2]$ with $\epsilon_1$ the smallest $\epsilon$ value for which $\lambda_{K+1}$ is non-increasing and $\epsilon_2$ the smallest value for which $\lambda_1$ is non-decreasing. For this particular data set, the CLMR range is approximately $[100, 300]$ for $K > 1$ ($K$ is an upper bound on the intrinsic dimension $d$ of the data). Hence, the CLMR method would choose an $\hat{\epsilon}$ of at least 100 (200 if the middle of the CLMR interval is used).

(a)  (b)

Figure 1: Dataset g241d. (a) costs $\hat{D}$ for one sample of $N' = 200$ and $d' = 1, 2, 4\ 8$, showing pronounced minimum at $\hat{\epsilon} = 53.1$ for $d' = 1$ (the lowest curve) and a weaker minimum for $d' = 2$; the range of $\epsilon$ searched was $[24, 482]$ (b) the nine largest singular values of local SVD versus $\epsilon$. We do not know the intrinsic dimension of these high-dimensional data. The figure shows why using a low dimensional projection, e.g. $d' = 1$ may be a practical strategy. One sees also that chosing $\epsilon$ by the CLMR will result in values of at least $100 - 300$, depending which parameter $K > 1$ is chosen. The value chosen by crossvalidation is 54.

### 1.2  Experiments with smoothing

Figure 2 shows the results of the experiments with smoothing wihin the main paper for additional noise amplitudes.

**1.3**

This behavior is largely due to what happens at large values of $\epsilon$. At these values, the geometry converges to the degenerate case of the single point, for which $||g|| \to 0$ (there are no longer any distances to be measured). This means that, when $g_{\mathbb{R}^r}|_T\mathcal{M}$ is compared to $g$, the result is simply the 0 matrix minus the identity matrix, which is just the norm of the identity matrix in $d$ dimensions. Therefore, the distortion converges to a small value as $\epsilon \to \infty$, and for finite samples, this value may be even smaller than the one resulting from the use of the optimal $\epsilon$. In contrast, when $||g|| \to 0$, the dual metric $||h|| \to \infty$, so the computed distortion from $g_{\mathbb{R}^r}|T\mathcal{M}$ is going to be very high even for finite samples.

**1.4  SDSS Data Embedding**

The data consists of spectra of galaxies from the Sloan Digital Sky Survey. We extracted a subset of spectra whose signal-to-noise-ratio was sufficiently high, known as the *main sample*.

This set contains 675,000 galaxies observed in $D = 3750$ spectral bins. The data were pre-processed by first moving them to a common rest-frame wavelength and then filling-in missing data using weighted PCA.

In figure 3 we display a three-dimensional embedding of the main sample of galaxy spectra from the Sloan Digital Sky Survey. Colors in the above figure indicate the strength of Hydrogen alpha emission, a very nonlinear feature which requires dozens of dimensions to be captured in a linear embedding. The continuous variation of this feature is also indication of a smooth embedding. Additionally in figure 4 we display a region of this embedding along with the estimated Riemannian metrics at a subset of the points. The continuity of the Riemannian metrics across the embedding are evidence of a smooth embedding.

Figure 2: Distortions between embedding of noisy and of noiselss manifold data, for various $\epsilon$ values sample sizes $n$, and noise levels $\sigma$. The manifold is the `hourglass`, embedding in 3D by Laplacian Eigenmap, data in 13 dimensions; $\epsilon$ is the scale for the noisy data embedding, and the distortion shown is the lowest over all $\epsilon^*$ values for the noiseless data embedding; there were 5 replications in each experiment. The vertical lines are the same $\hat{\epsilon}$ from Figure 1 in the paper (10 replicates). One sees that $\hat{\epsilon}$ underestimates the minimum of the distortion, but is in the ball park. Note the large interval of small distortion and the comparatively small variance of $\hat{\epsilon}$.

Figure 3: SDSS galaxy embedding with hydrogen alpha.

Spectra Data -- before relaxation

Figure 4: A portion of the embedding is displayed along with the estimated Riemannian Metrics at a subset of the points. .