[Reviews · NeurIPS 2017]

Reviewer 1



This paper proposes a method for improving the graph Laplacian by choosing the bandwidth parameter in the heat kernel for the neighborhood graph. The method uses geometric consistency in the sense that the Riemannian metric estimated with the graph Laplacian should be close to another estimate of the Riemannian metric based on the estimation of the tangent space. Some experiments with synthetic and real data demonstrate promising results of the proposed method. The proposed method for choosing bandwidth seems novel and interesting. It is based on the very natural geometric idea for capturing the Riemannian geometry of the manifold. There are, however, some concerns also as described below. - From the experimental results in Table 1, it is not very clear whether the proposed method outperforms the other methods. In comparison with Rec, the advantage of GC are often minor, and GC^{-1} sometimes gives worse results. More careful comparison including statistical significance is necessary. - The authors say that d=1 is a good choice in the method. The arguments to support this claim is not convincing, however. It is not clear whether Proposition 3.1 tells that d=1 is sufficient. More detailed explanations and discussions would be preferable. - The paper is easy to read, but not necessarily well written. There are many typos in the paper. To list a few, -- l.36: methods are duplicated. -- eq.(2): it should be minus in the exponential. -- l. 179: on is duplicated. -- Caption of Figure 1: have are.

Reviewer 2



General comment: The idea of optimizing epsilon parameter by comparing the metric induced from Laplacian and the geometry (tangent space) determined without epsilon, is interesting. The approach for realizing this idea is well thought-out. Concerns: The empirical result that the working dimension d' can be small, (even one-dimension is nice), is somewhat surprising. The proposition 3.1 in this paper partly explain the reason, but it is not enough. It is nice if the author(s) provide theoretical guarantee that considering small dimensional submanifold is enough. The proof of Proposition 2.1, which gives a method to calculate cometric should be presented (in supplementary material). eq.(2), the minus "-" is missing from exponential function. The presentation of the experimental results should be improved. Particularly, it is hard to read Fig. 1.